# MEGClass: Extremely Weakly Supervised Text Classification via Mutually-Enhancing Text Granularities

**Priyanka Kargupta[1], Tanay Komarlu[1], Susik Yoon[1], Xuan Wang[2], Jiawei Han[1]**
[1]Department of Computer Science, University of Illinois at Urbana-Champaign
[2]Department of Computer Science at Virginia Tech
{pk36, tkomarlu2, susik, hanj}@illinois.edu,
xuanw@vt.edu

## Abstract

Text classification is essential for organizing unstructured text. Traditional methods rely on human annotations or, more recently, a set of class seed words for supervision, which can be costly, particularly for specialized or emerging domains. To address this, using class surface names alone as extremely weak supervision has been proposed. However, existing approaches treat different levels of text granularity (documents, sentences, or words) independently, disregarding inter-granularity class disagreements and the context identifiable exclusively through joint extraction. In order to tackle these issues, we introduce MEGClass, an extremely weakly supervised text classification method that leverages **M**utually-**E**nhancing Text **G**ranularities. MEGClass utilizes coarse- and fine-grained context signals obtained by jointly considering a document's most class-indicative words and sentences. This approach enables the learning of a contextualized document representation that captures the most discriminative class indicators. By preserving the heterogeneity of potential classes, MEGClass can select the most informative class-indicative documents as iterative feedback to enhance the initial word-based class representations and ultimately fine-tune a pre-trained text classifier. Extensive experiments on seven benchmark datasets demonstrate that MEGClass outperforms other weakly and extremely weakly supervised methods.

## 1 Introduction

Text classification is a fundamental task in Natural Language Processing (NLP) that enables automatic labeling of massive text corpora, which is necessary in many downstream applications (Rajpurkar et al., 2016; Zhang et al., 2022b; Tang et al., 2015). Prior works train text classifiers in a fully-supervised manner (Yang et al., 2016, 2019; Zhang et al., 2015) that requires a substantial amount of training data, which is expensive and time-consuming, especially in emerging domains that require the su-

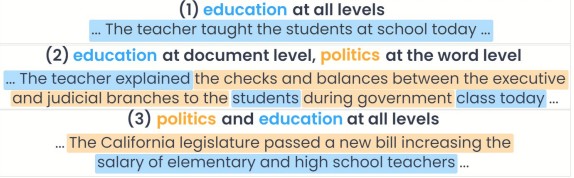

Figure 1: These are the three document types featured within a corpus. While existing methods can only distinguish between (1) and (3), MEGClass addresses all three types as well as minimizing (3) from the constructed pseudo-training dataset.

pervision of domain experts. Recent works have explored weakly supervised text classification, where a few labeled documents or class seeds are provided to serve as weak supervision (Meng et al., 2018; Mekala and Shang, 2020; Agichtein and Gravano, 2000; Tao et al., 2018; Zeng et al., 2022). These works typically compile a pseudo-training dataset from a given corpus by assigning a pseudo-label to each document based on its alignment to a specific class. The pseudo-training dataset aims to be a confident substitution for an expert-curated collection of class-indicative documents, ultimately being used to fine-tune a text classifier.

The motivation of this work begins with the assumption that there are three types of documents with a ground truth class topic, as illustrated in Figure 1: documents discussing (1) the class topic at the word, sentence, and document level, (2) the class topic at the document-level but with multiple other topics mentioned at the word or sentence level, or (3) multiple topics as well as the class topic at all levels. Existing weakly supervised methods focus on word-level (Meng et al., 2018, 2020b; Mekala and Shang, 2020) or document-level (Wang et al., 2021) information independently and only ever associate one pseudo-label to a document, which may allow them to differentiate documents of type (1) from those of type (3). However, this notion is fundamentally *unrealistic* for real-world data, which often takes on the inter-

granularity class disagreements exhibited in type (2), and hence encounters the risk of the word- and sentence-level topics overriding the true document-level class. For example, the *education*-news article of type (2) uses strong *political* terms to discuss the government being the subject matter of a teacher's lesson to students, which may mistakenly lead it being classified as "politics" instead of "education". In general, documents may also feature several indeterminate sentences that do not confidently indicate any sort of class alignment.

The existing classification methods unfortunately do not address such inconsistency concerns among different levels of granularity. Several approaches (Meng et al., 2018, 2020b) generate pseudo-labels based on the individual words or sentences instead of the entire document and evaluate their confidence in aligning to a single class. This may lead to two glaring issues: (1) many words and sentences are only considered high quality and class-indicative based on the context of their parent document, and (2) they could be representative of classes *different from the document's fundamental topic* (e.g., "checks and balances" and "judicial branches" in conflict with "education" in Figure 1). On the other hand, another approach (Wang et al., 2021) heavily relies on both word-level and document-level, but independently– resulting in both overconfident and vague document representations used for generating pseudo-labels.

In this study, we attack this problem by utilizing all three levels of text granularity (words, sentences, and documents) in a manner that allows them to mutually enhance each other's significance in understanding which class a given document aligns with. Furthermore, we consider a document's *class distribution* instead of a single class such that we rank documents with a narrower distribution higher and can correct any initial misclassifications (e.g., education > politics in Figure 1) through iterative feedback. Doing so will allow us to exploit three levels of insight into the document's local (word/sentence) and global (document) class distribution and jointly factor them into our understanding of the document and final pseudo-training dataset construction.

In this study, we propose a novel framework **MEGClass** to perform extremely weakly supervised text classification using *mutually enhancing text granularities*. Specifically, when only the class surface names are given, MEGClass first identi-

fies class-indicative keywords and uses them to represent classes and sentences. MEGClass then automatically weighs the importance of each sentence in classification for more accurately estimating a document's class distribution. MEGClass further leverages a multi-head attention network to learn a document representation that reflects the critical information at multiple text granularities. Finally, MEGClass takes the most confident *document* representations to enhance our initial *word-based* class representations through our iterative feedback approach, helping our model better understand what an e.g., "education" document should look like at all levels of granularity. Because our multi-granularity approach can handle all three document types when constructing the final pseudo-training dataset, we allow our fine-tuned final classifier to be more robust to challenging real-world documents. Comparing with existing weakly and extremely weakly supervised methods, MEGClass achieves a stronger performance on most datasets, especially with longer documents and fine-grained classes. Our contributions are summarized as follows:

1. To the best of our knowledge, this is the first work to exploit mutually enhancing text granularities for extremely weakly supervised text classification (only given the class label names).

2. We propose a novel method, MEGClass, which produces a quality pseudo-training dataset through class distribution estimation and contextualized document embeddings, refined by iterative feedback. The pseudo-training dataset is used to fine-tune a text classifier.

3. Experiments on seven datasets demonstrate that MEGClass outperforms existing weakly and extremely weakly supervised methods, significantly in long-document datasets and competitively in shorter, coarse-grained datasets.

**Reproducibility:** We release our data and source code[1] to facilitate further studies.

## 2 Related Work

### 2.1 Weakly Supervised Text Classification

In real world applications, the acquisition of human annotations from domain experts is time-

---

[1] https://github.com/pkargupta/MEGClass

consuming and expensive. Thus, prior methods leverage supervision from existing large knowledge bases such as Wikipedia to serve as distant supervision (Chang et al., 2008; Song and Roth, 2014; Gabrilovich and Markovitch, 2007), while other works utilize heuristic rules (Badene et al., 2019; Ratner et al., 2016; Shu et al., 2020) or seed keywords (Meng et al., 2018; Mekala and Shang, 2020; Agichtein and Gravano, 2000; Tao et al., 2018). While they mitigate significant human labeling efforts, it is often impractical to assume that a large knowledge base is available.

## 2.2 Extremely Weakly Supervised Text Classification

Recently, methods exploring the extremely weakly supervised setting are achieving motivating results. These methods exploit only the provided class label names to discover class-indicative keywords and generate pseudo-labels for classifier training. LOTClass (Meng et al., 2020b) utilizes pre-trained language models (PLMs) as a linguistic knowledge base to derive class-indicative keywords. Similarly, ConWea (Mekala and Shang, 2020) leverages contextualized representations of word occurrences and seed words to disambiguate class keywords. X-Class (Wang et al., 2021) discovers keywords to construct static class & document representations which are clustered to generate pseudo-labels. ClassKG (Zhang et al., 2021) leverages a Graph Neural Network (GNN) to iteratively explore keyword-keyword correlation for each class. WDDC (Zeng et al., 2022) utilizes cloze style prompts to obtain words that summarize the document content. They propose a latent variable model to learn a word distribution that maps to the predefined categories as well as a document classifier simultaneously. NPPrompt (Zhao et al., 2022) proposes to fine-tune PLMs on related words obtained through embedding similarities. This fine-tuned PLM is leveraged to prompt candidate labels which are aggregated into semantically similar labels for the final document prediction. Finally, PIEClass (Zhang et al., 2023) is a concurrent work that utilizes zero-shot prompting of PLMs in order to get pseudo labels based on contextualized text understanding. However, all of these methods fail to account for intergranular class disagreements and place full confidence in the initially chosen target class. Furthermore, these methods ignore the rich information that can be derived from jointly considering multi-text granularities.

## 2.3 Exploiting Multi-Text Granularities

In addition to both word- and document-level granularities being utilized in weakly supervised text classification, other tasks such as topic mining, story discovery, and document summarization have exploited various text granularity levels. Recent studies on topic mining utilize PLMs by employing context-aware word representations and sentence-level context (Zhang et al., 2022a). Furthermore, recent studies for online story discovery utilize pretrained sentence representations of articles along with thematic keywords (Yoon et al., 2023b) and continual self-supervision (Yoon et al., 2023c). For interpretable topic discovery, Top-Clus (Meng et al., 2022) leverage low-dimensional latent topics for attention-based embedding and clustering. PDSum (Yoon et al., 2023a) proposes a prototype-based continuous summarization of evolving multi-document set streams, by incorporating word- and sentence-level themes to extract representative sentences.

# 3 Framework

## 3.1 Preliminaries

**Problem Formulation.** We address an extremely weakly supervised text classification task, which expects an input of class names and a set of documents and outputs a single class label per document. Specifically, let document $d_i \in D = [s_1, s_2, \ldots, s_{|d_i|}]$ be a list of sentences, where we assume that $d_i$ primarily falls under a specific class $c_k$. Each class $c_k \in C$ has a surface name, which we utilize as initial insight into what the class is about. The set of documents with a ground truth class of $c_k$ is denoted as $C_k$. We also consider a sentence $s_j = [w_1, w_2, \ldots, w_{|s_j|}]$ as a list of words.

**Class Representations.** Weakly supervised text classification problems can be initially approached by getting class representations using a pre-trained language model. Previous works (Aharoni and Goldberg, 2020; Wang et al., 2021) find that using a simple and weighted average of contextualized word representations is effective for preserving the domain information of documents and hence effective for clustering relevant documents in each class. For instance, X-Class (Wang et al., 2021) estimates class representations by a harmonic mean of the $T$ static word representations (Ethayarajh, 2019) closest to the class surface name.

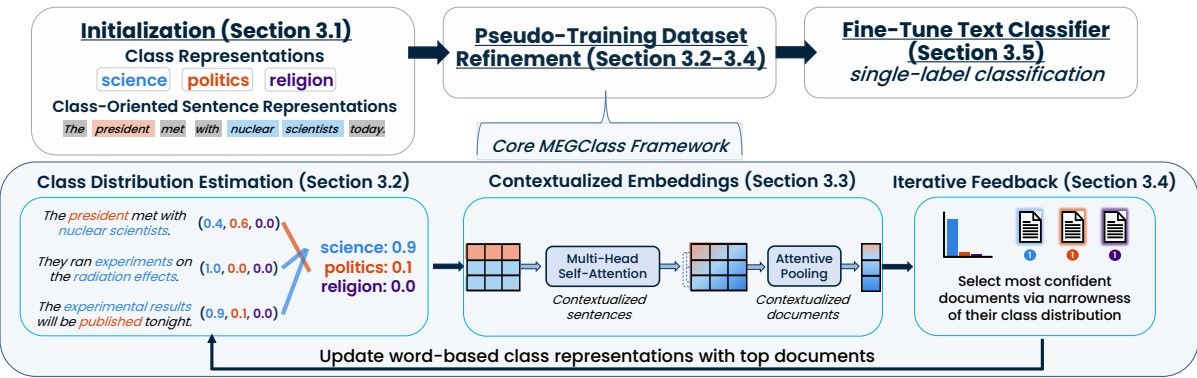

Figure 2: We propose MEGClass, which given a raw text input corpus and user-specified class names, learns contextualized document representations that reflect the most discriminative indicators of its potential class using a regularized contrastive loss. It then utilizes the most confident documents for each class to update its understanding of the respective class, ultimately constructing a pseudo dataset to fine-tune a text classifier.

**Class-Oriented Sentence Representations.** Each word in a sentence is assigned a weight based on the similarity between its closest class and the average of all class representations, finally to derive a class-oriented sentence representation. Any class (Meng et al., 2020a; Zhang et al., 2022a) or sentence representations (Reimers and Gurevych, 2019) can be used for this purpose (explored further in Appendix A.3). For MEGClass, we use the initialization introduced in X-Class's (Wang et al., 2021), but we do so based on only the sentence-level context and not the document-level context as X-Class does.

Figure 2 shows the overall framework of MEGClass, which exploits these class and sentence representations to learn contextualized document representations along with varying class distributions, to derive a pseudo dataset to fine-tune a classifier.

## 3.2 Class Distribution Estimation

In order to learn the most critical indicators of a document's potential true class, we must first identify which classes are viable candidates. To provide flexibility and correct any initial misclassifications, we choose to represent a document's set of class candidates as a distribution as opposed to a binary label for each class. A narrower class distribution indicates higher confidence in assigning a class label to the given document. A naive method to determine the target class distribution is to average a document's sentence representations and compute the cosine-similarity between the average and each class representation. This unrealistically assumes that all sentences have equal significance to a document's underlying context. Consequently, given our task, we claim that a sentence's significance correlates with how informative it is in indicating a single distinct class. Hence, for classifying sports, sentences that can represent two or more classes (e.g. "the player got the ball") are vague and should be weighed less than a sentence related to a single class (e.g. "he hit a home run", which is only relevant to baseball).

We quantify this by noting sentence $s_j$'s cosine-similarity to its top two classes individually ($q_j^0$ and $q_j^1$ respectively). The larger the gap ($q_j^0 - q_j^1$) is between the two top cosine-similarities, the more discriminative $s_j$ is to its top class. For example, "he hit a homerun" would be significantly closer to baseball than tennis and hence would have a larger class gap than "the player got the ball".

In addition to measuring class discriminativeness, we must also consider that a sentence $s_j$'s significance is relative to the other sentences in the document $d_i$, and that the sum of all a document's sentence weights must equal to 1. Thus, we choose to divide each sentence's class gap by the sum of all sentence class gaps within document $d_i$:

$$s_{j,weight} = \frac{q_j^0 - q_j^1}{\sum_{l=1}^{|d_i|}(q_l^0 - q_l^1)} \tag{1}$$

This allows us to reduce the impact that vague sentences (close to multiple classes) have on the target class distribution. In Figure 4, we demonstrate the superior performance of this metric (6% greater) compared to other techniques.

Once we have the confidence of each sentence $s_j$ within the document $d_i$, we perform a weighted label ensemble where each sentence will vote for its most similar class $c_k$ with its vote weight $s_{j,weight}$, and the voting results are aggregated into a class distribution $P(d_i \in C_k)$:

$$P(d_i \in C_k) = \sum_{\substack{s_j \in d_i \\ q_j^0 = \cos(s_j, c_k)}} s_{j,weight} \qquad (2)$$

### 3.3 Contextualized Representations

**Contextualized Sentence Representations.** As we discuss in Section 3.1, the sentence representations are computed independently from the other sentences within a document. However, sentences that seem vague in isolation (e.g., "he walked to the podium") may become significant when key information from other sentences is introduced (e.g., "the president held a press conference today"). Hence, we aim to incorporate the pair-wise relationships between the sentences $s_j$ in a document $d_i$. We accomplish this by learning contextualized sentence representations $cs_j$ using a multi-head self-attention mechanism (Vaswani et al., 2017):

$$\begin{aligned} CS(d_i) &= [cs_1, cs_2, \ldots, cs_{|d_i|}] \in \mathbb{R}^{|d_i| \times h_{cs}} \\ &= l_{ln}(l_{mhs}([\mathbf{E}_j | s_j \in d_i]) + [\mathbf{E}_j | s_j \in d_i])) \end{aligned} \qquad (3)$$

where $h_{cs}$ is the number of hidden dimensions, $[\mathbf{E}_j | s_j \in d_i]$ denotes the initial class-oriented sentence representations for document $d_i$, $l_{mhs}$ is a multi-head self attention layer, and $l_{ln}$ a feed forward layer with layer normalization.

By computing contextualized sentence representations, we incorporate both the word-level (via class-oriented weights as discussed in Section 3.1) and the document-level information (via multi-head self-attention). However, in order to construct a contextualized document representation, we must identify which sentences contribute the most towards the document's fundamental context.

**Contextualized Document Representations.** Unlike Section 3.2, where we had to determine each sentence's significance based on its class-discriminativeness, we now have a target class distribution and document-level context integrated into our sentence representations. Recent document summarization methods show that learning sentence-level attention weights in representing documents can help highlight documents' theme (Yoon et al., 2023a). Similarly, in our case, we attentively combine the contextualized sentences to represent their parent document, while promoting the document's underlying theme to be akin to the target class distribution that we have computed.

Specifically, we utilize attentive pooling to learn attention weights for each of the contextualized

sentences $cs_j$, indicative of a sentence's relative significance to its parent document $d_i$:

$$\begin{aligned} CD(d_i) = cd_i &= \sum_{j=1}^{|d_i|} \alpha_j cs_j \in \mathbb{R}^{h_{cd}} \\ &= \sum_{j=1}^{|d_i|} \frac{e^{(l_\alpha(cs_j))}}{\sum_{k=1}^{|d_i|} e^{(l_\alpha(cs_k))}} cs_j \end{aligned} \qquad (4)$$

where $h_{cd}$ is the number of hidden dimensions, $\alpha_j$ is an attention weight indicating the relative importance of $cs_j$ for representing $cd_i$, derived by an attention layer $l_\alpha(cs_j) = \tanh(W cs_j + b)V$ with learnable weights $W$, $b$, and $V$.

Then, we minimize a weighted regularized contrastive loss (Chen et al., 2020) to ensure that the contextualized document representations $cd_i$ and sentence attention weights are updated to reflect all viable class candidates $c_k$, relative to the target class distribution:

$$\begin{aligned} \mathcal{L}_{WCon}(d_i, C, P(d_i)) &= -\sum_{k=1}^{|C|} P(d_i \in C_k) \\ &\times \log \frac{e^{(\cos(\mathbf{cd}_i, \mathbf{c}_k))/\tau}}{\sum_{c_n \in C} e^{(\cos(\mathbf{cd}_i, \mathbf{c}_n)/\tau)}} \end{aligned} \qquad (5)$$

where the temperature scaling value is $\tau$, and the target class distribution is $P(d_i \in C_k)$. Note that this regularized contrastive loss is designed specifically for MEGClass as unlike previous methods (Mekala and Shang, 2020; Meng et al., 2020b; Wang et al., 2021; Zhang et al., 2021; Yoon et al., 2023a,c), we do not artificially fit a contextualized document representation to a single class without the confidence to support it. By retaining the class candidates within the contextualized representation itself, we can also determine which documents are the most confidently single-topic.

### 3.4 Iterative Feedback

Thus far, we have been using the initial word-based class representations (Section 3.1) as our reference for a target class. However, now that we can identify the documents with the narrowest class distribution and hence highest class-confidence, we can utilize them to enhance their corresponding word-based class representations. Hence, the iterative feedback will provide more insight into discovering new potential class indicators within the other less confident documents.

**Updating the Class Set.** Following the application of principal component analysis (PCA) to

PLM-based text clustering seen in (Aharoni and Goldberg, 2020), we first reduce any potential redundant noise in both our contextualized document and class representations using PCA. Each transformed representation will then have its closest class as its pseudo-label and a confidence score equal to the corresponding cosine-similarity. For each class $c_k$, we rank the documents (that have pseudo-label $c_k$) based on their confidence and add the top $k\%$ documents to each class's set, where each set already contains the initial word-based class representation. For each following iteration, the documents are re-ranked, and the class sets are consequently recomputed.

In order to utilize this feedback, we replace the initial class representations with the average of each class set and use the learned contextualized sentence representations $cs_j^i$ and corresponding sentence weights $\alpha_j$ for recomputing the target class distribution (Equation 2). This allows us to incorporate both document-level and sentence-level feedback throughout our iterations. We avoid word-level feedback in order to mitigate the efficiency issues seen in previous works such as (Zhang et al., 2021) and (Zeng et al., 2022). Overall, we find that using an updated class distribution for re-learning new contextualized representations significantly boosts performance (as shown in our experiments section) and avoids significant error propagation (more details in Appendix A.6).

### 3.5 Text Classifier Fine-Tuning

Unlike prior methods, MEGClass does not assume that a document will exclusively map to a single class. Specifically, we find that retaining a document's class distribution information is crucial to identifying documents that discriminatively align with a single class (e.g. a 90% politics, 10% health article is a stronger candidate than a 70% politics, 30% health article) for our pseudo-training dataset.

Using the confidence score as described in Section 3.4, we select the top $\delta\%$ of the documents for each class. These documents and their pseudo-labels are used to fine-tune a pre-trained text classifier as ground-truth in order for our method to be generalized and applied to unseen documents (Appendix A.2). This is an established and studied noisy training scenario (Goldberger and Ben-Reuven, 2017; Angluin and Laird, 1988). Thus, since we know how confident we are on each document, we can select the most confident documents

to train our text classifier (Devlin et al., 2019).

## 4 Experiments

### 4.1 Experimental Setup

**Datasets.** We use seven publicly available benchmark datasets to provide a wide range of label granularity, domain, and document length. We covered sentiment-based reviews (Yelp), lengthy news articles (20News, NYT-Topic, NYT-Location), concise summaries with abstract classes (Books), and fine-grained classes (NYT-Fine, 20News-Fine). Table 4 in the Appendix specifies the label names used for each dataset:
(1) **Yelp** (Zhang et al., 2015) is for sentiment polarity classification of Yelp business reviews, adapted from the Yelp Dataset Challenge in 2015. (2) **Books** (Wan and McAuley, 2018) is a dataset of book titles and descriptions used for book genre categorization, collected from a popular book review website Goodreads. (3) **20News** (Lang, 1995) is a long-document dataset collected from 20 different newsgroups for topic categorization. (4) **NYT-Topic** (Meng et al., 2020a) is a long-document dataset collected from the New York Times for topic categorization. (5) **NYT-Loc** (Meng et al., 2020a) uses the same corpus as NYT-Topic, collected from the New York Times, but is used for location categorization. (6) **NYT-Fine** (Wang et al., 2021) is a long-document dataset collected from the New York Times with 26 fine-grained classes (e.g., surveillance, the affordable care act). (7) **20News-Fine** (Lang, 1995) is a dataset collected in the same manner as 20News, but is partitioned into 20 fine-grained classes (e.g., graphics, windows, baseball).

| Dataset | Domain | Granularity | Classes | Docs |
|---|---|---|---|---|
| Yelp | Sentiment | Coarse | 2 | 38,000 |
| 20News | News | Coarse | 5 | 17,871 |
| NYT-Topic | News | Coarse | 9 | 31,997 |
| NYT-Loc | News | Coarse | 10 | 31,997 |
| Books | Goodreads | Coarse | 8 | 33,594 |
| NYT-Fine | News | Fine | 26 | 13,081 |
| 20News-Fine | News | Fine | 20 | 4,792 |

Table 2: *Dataset Statistics.*

**Compared Methods.** We compare MEGClass to the following six methods that have shown the most promising results thus far. We use Micro-/Macro-F1 as our evaluation metrics (Appendix A.9). More details of the parameter and supervision settings of each model can be found in Appendix A.7. (1) **NPPrompt** (Zhao et al., 2022) collects related

| Model | Yelp | 20News | NYT-Topic | NYT-Loc | Books | NYT-Fine | 20News-Fine |
|---|---|---|---|---|---|---|---|
| Supervised | 95.7/95.7 | 96.60/96.60 | 95.98/95.01 | 96.0/95.0 | 81.0/81.0 | 98.0/96.6 | 96.39/96.36 |
| ConWea | 71.4/71.2 | 75.73/73.26 | 81.67/71.54$^\dagger$ | 85.31/83.81 | 52.3/52.6 | 76.23/69.82 | 48.7/48.7 |
| LOTClass | 87.4/87.2 | 73.78/72.53 | 67.11/43.38 | 58.49/58.96 | 19.9/16.1 | 15.0/20.21 | 9.4/9.6 |
| X-Class | 86.8/86.8 | 73.17/73.07 | 79.01/68.62 | 89.51/89.68$^\dagger$ | 53.6/54.2 | 85.68/67.36 | 58.7/58.5$^\dagger$ |
| ClassKG | **91.2/91.2** | 81.0/82.0$^\dagger$ | 72.06/65.76 | 86.84/83.35 | 55.0/54.7$^\dagger$ | 88.86/70.5$^\dagger$ | 52.29/52.1 |
| WDDC-MLM | 81.2/81.1 | 81.21/68.82 | 81.5/69.2 | 88.84/86.91 | 53.86/53.75 | 87.41/68.34 | 51.1/50.2 |
| NPPrompt | 81.2/81.1 | 68.9/68.8 | 64.6/64.2 | 53.9/53.8 | 49.6/49.7 | 55.2/54.9 | 48.6/48.3 |
| **MEGClass** | 87.41/87.41$^\dagger$ | **81.72/80.63** | **85.42/68.03** | **93.06/91.93** | **56.35/55.71** | **89.24/71.06** | **66.37/64.24** |
| MEG-Init | 78.17/77.44 | 75.93/74.57 | 79.85/64.77 | 62.84/65.57 | 50.28/50.51 | 82.48/66.74 | 52.30/54.02 |
| MEG-CX | 84.24/84.24 | 76.90/75.26 | 78.71/65.11 | 82.31/80.63 | 52.39/51.21 | 86.54/69.46 | 64.23/63.21 |

Table 1: *Evaluations of Compared Methods and MEGClass.* Our results are averaged over five trials with the variance provided as well. Both micro-/macro $F_1$ scores are reported due to imbalanced datasets. Supervised performance provides an upper bound. $\dagger$ denotes the second-best method.

words through embedding similarities obtained from a pre-trained language model. These related words are leveraged as labels to prompt a generative language model and aggregated as the matching result. (2) **WDDC-MLM (Zeng et al., 2022)** utilizes cloze style prompts to obtain words summarizing the document. They learn a latent variable model to map a word distribution to the predefined categories as well as a document classifier simultaneously. (3) **ClassKG (Zhang et al., 2021)** constructs a keyword graph with co-occurrence relations and self-trains a sub-graph annotator to generate pseudo labels for text classifier training. The class predictions iteratively update the class keywords. (4) **X-Class (Wang et al., 2021)** leverages BERT to generate class-oriented document representations. The document-class pairs are formed by clustering and used to fine-tune a text classifier. (5) **LOTClass (Meng et al., 2020b)** constructs a category vocabulary for each class, using a PLM fine-tuned using a self-training and soft-labeling strategy. (6) **ConWea (Mekala and Shang, 2020)** uses a PLM to obtain contextualized representations of keywords. It then trains a text classifier and expands seed words iteratively. (7) **Supervised (Devlin et al., 2019)** fine-tunes a BERT text classifier on a task-specific training dataset with an 80/20 train-test split.

We denote our method as **MEGClass**; our experimental settings and hyperparameter sensitivity analyses are included in Appendix A.7 and 4.4.

We run a set of ablation studies in order to better understand the impact that each one of our modules has on the overall methodology. We have two ablation versions. **MEG-Init** refers to the labels obtained when we select the class with the maximum weight in the initial target class distribution as specified in Equation 2. **MEG-CX** refers to the labels obtained in the first iteration when we select

the class with the highest cosine similarity between the transformed contextualized document representations and the transformed class representations, as discussed in beginning of Section 3.4.

## 4.2 Overall Performance Results

Table 1 shows our results on the Yelp, 20News, NYT, and Books datasets. Overall, MEGClass performs the best on the long text coarse-grained datasets. However, MEGClass trails ClassKG on the Yelp dataset while being competitive with other weakly supervised approaches. It is important to note that Yelp contains shorter text documents when compared against News datasets due to the nature of Yelp reviews, which does not benefit from the varying levels of text granularity as much as longer documents. In addition, ClassKG has a drastically longer run time as demonstrated in Figure 3. Finally, we show that MEGClass significantly beats the other datasets on 20News-Fine and NYT-Topic micro-f1 and approaches fully supervised performance on the NYT-Loc dataset (first row of Table 1). We expand upon these results through a case study in Appendix A.4 and A.5.

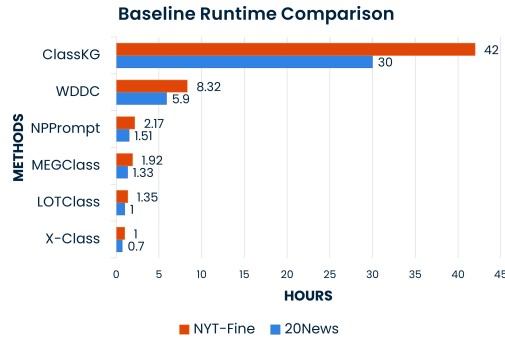

Figure 3: Running time of compared methods.

**Fine-Grained Classification.** In Table 1, we also examine MEGClass's performance on fine-grained text classification on the NYT-Fine and 20News-

Fine datasets, which have annotations for 26 and 20 fine-grained classes respectively. We compare our method against other state-of-the-art weakly supervised methods and show that MEGClass performs the best. We largely attribute this performance to longer text documents and the fine-grained label space containing highly distinct classes. With the expanded keywords, our contrastive learning process leverages the distinct label space to learn the attributes that are common between data classes and attributes that set apart a data class from another.

**Ablation Studies.** In our thorough evaluation, we analyzed the relative contributions of each component and found that the impact of our learned contextualized representations > iterative feedback > class distribution estimation. Specifically, the quality of the joint representations that we learn determines which documents we deem as sufficiently confident to enhance the class representations as iterative feedback, which then heavily influences the accuracy of the estimated class distribution used for the next iteration. We can see this through the immediate boost in performance between MEG-Init and MEG-CX in Table 1 in Figure 6, as well as the overall high quality documents chosen across all different datasets, shown in Table 10.

### 4.3 Weighted Label Ensemble Analysis

In Figure 4, we compare different mechanisms for computing the sentence weights used in the weighted label ensemble (Section 3.2). We compare using equal sentence weights, sentence centrality (Liang et al., 2021), and class discriminativeness (what MEGClass proposes in Equation 1). Sentence centrality is a popular sentence weight metric in document summarization (Liang et al., 2021), measuring the similarity between a given sentence at position $i$ and all sentences with positions $\geq i + 1$. Its values tend to reflect leading-sentence bias. Figure 4 demonstrates that the class discriminativeness metric we propose performs the best overall, especially on long-document datasets like NYT where the leading sentences do not reflect all of the main topics. This demonstrates the need for a class-discriminative weighted label ensemble.

**Iterative Feedback Analysis.** The top graphs in Figure 5 demonstrate the significant benefit of utilizing the top-$k\%$ class-indicative documents as feedback for updating the target class distribution. Furthermore, after the first iteration, the top

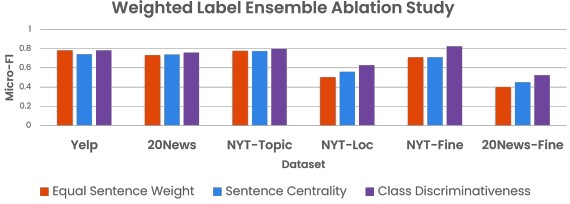

Figure 4: Effects of sentence weight computation metric during iteration #1 for weighted label ensemble proposed in Equation 1.

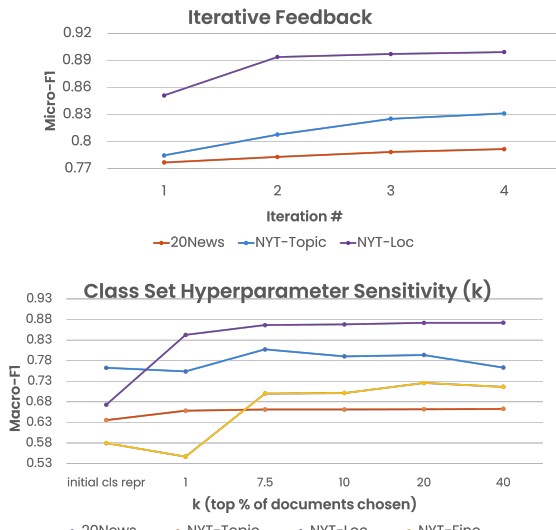

Figure 5: **Top.** Micro-F1 scores after transforming contextualized document representations over 4 iterations. **Bottom.** Sensitivity of top $k\%$ documents chosen to update class set as iterative feedback. "Initial cls repr" denotes the initial class word-based representations.

7.5% of documents chosen have an average micro-/macro F1 score of 90.75/86.93 across all datasets, including Books (see Table 10 in Appendix A.6 for the full results).

We expand upon these results by showing the qualitative (Table 3) quality of the set of class-indicative documents. We can see that relative to the overall difficulty of the dataset, the accuracy of documents chosen is high across all classes (as seen through the macro-F1), and when we observe the highest-ranked document for each 20News class, the document is very representative of the class—even after the first iteration.

### 4.4 Hyperparameter Sensitivity

We introduce four core hyperparameters in MEG-Class, and we conduct hyperparameter sensitivity analyses for each one. At the high-level, our framework consists of three main parts: (1) initialization (Section 3.1), (2) pseudo-training dataset re-

| Class | Document Excerpt |
|-------|------------------|
| Science | [...] isotope changes don't normally affect chemistry, a consumption of only heavy water would be fatal, and that seeds watered only with heavy water do not sprout. Does anyone know about this? I also heard this. The reason is that deuterium does not have exactly the same reaction rates as hydrogen due to its extra mass (which causes lower velocity, Boltzmann constant, mumble). [...] |
| Politics | [...] it had a bit part in the much larger political agenda of President Clinton. [...] NJ assembly votes to overturn assault weapon ban. Feb 28th - Compound in Waco attacked. On Feb. 25th the New Jersey assembly voted to overturn the assault weapon ban in that state. It looked like it might be a tight vote, but the Senate in N.J. was going to vote to overturn the ban. [...] |
| Religion | [...] the Oriental Orthodox and Eastern Orthodox did sign a common statement of Christology, in which the heresy of Monophysitism was condemned. So the Coptic Orthodox Church does not believe in Monophysitism. Sorry! What does the Coptic Church believe about the will and energy of Christ? [...] |

Table 3: *Class Set Examples.* The highest-ranked document after one iteration in three of the 20News top-$k\%$ class sets (other two classes are comparable in quality).

finement (Sections 3.2-3.4), and (3) classifier fine-tuning (Section 3.5). The second part is our main component, which involves the following three sub-steps:

1. *Class distribution estimation* (Section 3.2) → there are no hyperparameters for estimating the initial class distribution of each document.

2. *Contextualized embeddings* (Section 3.3) → for learning document representations that jointly integrate sentence and document level information, the hyperparameters we introduce are the training epochs and regularization temperature used in the weighted regularized contrastive loss. Specifically, in Figure 6 we find that MEGClass's performance remains stable for temperature $\tau$ values below 0.3. We also demonstrate in Figure 7 that increasing the number of epochs does not benefit MEGClass significantly. In fact, MEGClass starts to stagnate after Epoch 3 and is more dependent on the top k% of documents selected.

3. *Iterative feedback for refining the first two sub-steps* (Section 3.4) → we introduce the following hyperparameters: (1) $k$ for choosing the top $k\%$ contextualized document representations to enhance each class representation. (2) the number of iterations of feedback. Figure 5 presents a sensitivity analysis on both of these, and shows that we reach a relatively stable set of confident documents at $k = 7.5\%$ that significantly enhances the quality of the initial keyword-based class sets across all seven datasets.

For a fair comparison with the baselines, we use the same hyperparameter settings for the initializa-

tion and classifier fine-tuning steps (e.g. for the final classifier fine-tuning, choosing the top % of documents as the pseudo-training dataset).

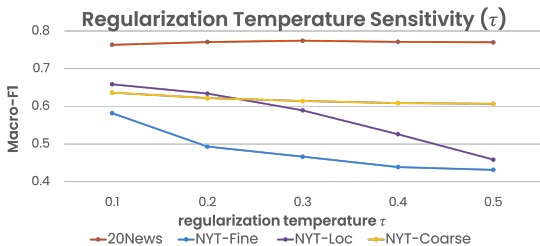

Figure 6: Sensitivity of the temperature scaling value for the weighted regularized contrastive loss. Macro-F1 scores are computed after the first iteration.

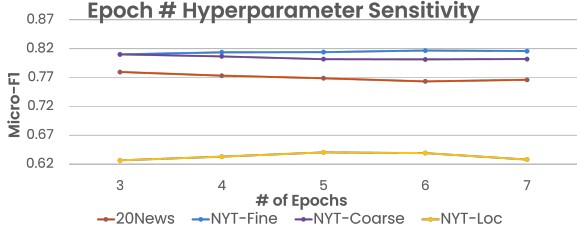

Figure 7: Sensitivity of the number of training epochs during the first iteration for learning the contextualized sentence and document representations.

## 5 Conclusion

In this work, we propose our novel method **MEGClass** for extremely weakly supervised text classification by exploiting mutually-enhancing text granularities. MEGClass learns new document representations that reflect the strongest class indicators with respect to all three levels of text granularities. MEGClass identifies the top documents with a narrower class distribution (more confidently single-topic) to use as feedback for the next iteration, and it ultimately constructs a pseudo-dataset of documents for fine-tuning a pre-trained text classifier. We demonstrate MEGClass's strong performance and stability through extensive experiments on varying domains (news and reviews) and label spaces (topics, locations, and sentiments). Due to MEGClass's novel use of mutually enhancing text granularities, further exploration can be done to utilize MEGClass for information extraction at varying text granularities. Furthermore, more sophisticated nonlinear dimensionality reduction methods can be a potential aspect to explore. Finally, handling class hierarchies may improve MEGClass's fine-grained classification abilities.

## Ethics Statement

Based on our current methodology and results, we do not expect any significant ethical concerns, given that text classification is a standard problem across NLP applications and basing it on extremely weak supervision helps as a barrier to any user-inputted biases. However, one minor factor to take into account is any hidden biases that exist within the pre-trained language models used as a result of any potentially biased data that they were trained on. We used these pre-trained language models for identifying semantical similarities between class names and documents/sentences and did not observe any concerning results, as it is a low-risk consideration for the domains that we studied.

## Limitations

In our experiments, we find that the expanded keywords can be vague and unrelated for topics described by a singular homonym. For example, the class label name "Cosmos" in the NYT-Fine dataset generates keywords related to the New York Cosmos rather than Astronomy. When we modify the label names by replacing homonyms with labels that provide surface-level context for each topic, we find that this increases MEGClass's macro-f1 performance by approximately 6.64 points (77.70), demonstrating that the selection of label names and seed words is critical to provide contextual information for topics. This information is especially crucial for fine-grained text classification as fine-grained classes like "international business" and "economy" can be indistinguishable to pre-trained language models without local or global context from sentences and documents.

## Acknowledgements

This research was supported in part by the US DARPA KAIROS Program No. FA8750-19-2-1004 and the National Research Foundation of Korea (Basic Science Research Program: 2021R1A6A3A14043765). Any opinions, findings, and conclusions or recommendations expressed herein are those of the authors and do not necessarily represent the views, either expressed or implied, of DARPA or the U.S. Government.

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

# A  Appendix

## A.1  Dataset Classes

Table 4 contains the label names that we use for each dataset.

## A.2  Robustness of Final Text Classifier

We conduct a performance comparison to verify the robustness of our fine-tuned pre-trained text classifier proposed in Section 3.5 against unseen documents, as shown in Table 5. Specifically, we

| Dataset | Class Words |
|---|---|
| Yelp | bad, good |
| 20News | computer, sports, science, politics, religion |
| NYT-Topic | business, politics, sports, health, education, estate, arts, science, technology |
| NYT-Loc | united_states, iraq, japan, china, britain, russia, germany, canada, france, italy |
| Books | children, comics_graphic, fantasy_paranormal, history_biography, mystery_thriller_crime, poetry, romance, young_adult |
| NYT-Fine | music, baseball, international_business, abortion, military, football, golf, television, economy, dance, soccer, cosmos, surveillance, law_enforcement, basketball, federal_budget, movies, stocks_and_bonds, gun_control, energy_companies, environment, hockey, the_affordable_care_act, immigration, tennis, gay_rights |
| 20News-Fine | atheism, graphics, microsoft, ibm, mac, motif, autos, motorcycles, baseball, hockey, encryption, electronics, medicine, space, christian, guns, arab |

Table 4: Label names used for each dataset

| Dataset | Train | Test |
|---|---|---|
| Yelp | 91.27/91.27 | 85.85/85.30 |
| 20News | 91.24/90.20 | 71.00/70.12 |
| NYT-Topic | 94.75/72.28 | 85.06/67.95 |
| NYT-Loc | 97.53/97.37 | 92.65/90.99 |
| Books | 66.24/66.47 | 43.64/41.61 |
| NYT-Fine | 93.49/78.00 | 85.96/63.02 |
| 20News-Fine | 76.39/72.77 | 57.28/56.77 |

Table 5: Micro-/Macro-F1 performance of fine-tuned text classifier on train and test documents.

use a train-test split of 50-50, where the 50% training dataset consists of the top $\delta = 50\%$ document pseudo-dataset (Section 3.5) and the remaining $50\%$ in the testing dataset are the lower-ranked documents unseen by the final text classifier. We can see that the pseudo-dataset which MEGClass constructs is quite robust to unseen documents, especially given that the test documents are explicitly not as single-topic confident (hence ranked in the bottom $50\%$ of documents).

## A.3  Varying Class and Sentence Representations

In this section, we demonstrate the MEGClass can be applied to any initialization of sentence and class representations. For clearly comparing the performance and overall quality of the different representations, we present the final classifier's performance in Table 6 when using the original class and class-oriented sentence representations (CO) versus SentenceTransformers-based class and sentence representations (ST) respectively (Reimers and Gurevych, 2019).

For computing the class and sentence representations using SentenceTransformers, we use the model `all-mpnet-base-v2` to encode a class phrase (e.g. "This is [class name]" for Yelp and Books; "This article is about [class name]" for 20News, NYT-Topic, NYT-Loc, and NYT-Fine)

| Dataset | CO | ST |
|---|---|---|
| Yelp | 87.41/87.41 | **90.60/90.58** |
| 20News | **81.72/80.63** | 69.12/66.44 |
| NYT-Topic | **85.42/68.03** | 82.70/61.78 |
| NYT-Loc | **93.06/91.93** | 56.48/65.45 |
| Books | **56.35/55.71** | 50.68/51.21 |
| NYT-Fine | 89.24/71.06 | **90.32/77.28** |
| 20News-Fine | **66.37/64.24** | 38.00/44.09 |

Table 6: Micro-/Macro-F1 final classifier performance comparison between using class-oriented representations (CO) and SentenceTransformers-based (ST) sentence and class representations on MEGClass's original framework and parameter settings.

| Iteration | United States | Iraq | Britain |
|---|---|---|---|
| 1 | **0.4344** | 0.1423 | 0.4233 |
| 2 | 0.3295 | 0.1954 | **0.4751** |
| 3 | 0.2133 | 0.1773 | **0.6095** |
| 4 | 0.0895 | 0.1424 | **0.7680** |

Table 7: Class Distribution Updates For Document 3. We only show the classes with non-zero probabilities.

to represent each class, as well as encoding each sentence individually.

From Table 6, we can see that each type of representation performs well on different datasets. Specifically, SentenceTransformers seems to surpass the original class-oriented representations on Yelp and NYT-Fine, notably datasets which have abstract and indistinct classes respectively (e.g. Yelp: good vs. bad; NYT-Fine: phrase class-names, overlap between classes). On the other hand, the class-oriented representations perform well on datasets with distinct and well-represented classes.

| Class | Class Words |
|---|---|
| United States | united_states, united, u, states, american, america, usa, americans, americanism, nation, country, world |
| Iraq | iraq, iraqi, baghdad, iraqis, baghdadis, iraqiya, baghdadi, kuwaiti, kuwait, iraqs, kuwaitis, saddam |
| China | china, chinese, beijing, beijingers, guanxi, guangxi, nanzhang, guangming |
| Britain | britain, british, england, uk, kingdom, london, londoners, britons, briton, anglo, brits, britian |
| Russia | russia, russian, russians, moscow, soviet, siberia, soviets, novgorod, rossiya, leningrad, siberian, muscovite |
| Germany | germany, german, germans, deutschland, berliner, berlin, berliners, germanys, prussia, prussian, saxony, bavaria |
| Canada | canada, canadian, canadians, quebec, quebecers, toronto, montreal, montrealers, quebecois, ontario, ottawa, manitoba |
| France | france, french, paris, parisian, frenchman, frenchmen, frenchwoman, francais, frenchness, bordeaux, parisians, francaise |
| Italy | italy, italian, italians, italiano, italia, italiana, milanese, milan, tuscany, veneto, nazionale, florentine |

Table 8: Class-Indicative Words. The top 12 (or all if number of critical words identified was smaller as mentioned in Section 3.1) class-indicative terms listed in-order of similarity for the NYT-Loc dataset.

## A.4 Case Study

In this section, we walk through the critical modules of the MEGClass framework by showing their respective intermediate results on the NYT-Loc dataset and their respective impact in determining a final classification for a document.

In Table 8, we list the top 12 class-indicative words (in order of descending rank) identified through the algorithm provided by XClass (Wang et al., 2021) for each class in NYT-Loc. In some cases, based on the term-extraction algorithm, certain classes may have a smaller number of terms identified. These terms are utilized for computing the initial word-based class representations.

Furthermore, in Table 9, we include three different documents which MEGClass classifies correctly and baseline methods, X-Class and ClassKG, misclassify, and MEGClass on the hand classifies correctly. From the three examples, we can see that the incorrect classes that X-Class and ClassKG choose are typically mentioned within the document either explicitly or implicitly through strong class-indicative terms. However, just as shown in Figure 1, in document type (2), we cannot solely trust the word-level context to be able to independently identify the document-level context. Hence, these word-level red herrings (e.g. "Iraq's weapons" in Document 3) distract both X-Class and ClassKG from the true document-level context (e.g. the recent legislative vote in the *British* Parliament and scandal with the prime minister).

In Table 7, we additionally show how the class distribution that we estimate using the weighted label ensemble (Section 3.2) updates over each iteration as we refine the class sets with confident documents. We can see how critical incorporating confidence documents as iterative feedback is for Document 3 (excerpt provided in Table 9), which initially has a slightly higher probability of incorrectly belonging to "Iraq" but is corrected to "Britain" in the second iteration.

## A.5 Comparison with ChatGPT[2]

We conduct a performance comparison between MEGClass and ChatGPT on the NYT-Topic and NYT-Fine datasets. For prompting ChatGPT, we use the following template in an effort to prevent ChatGPT from generating new class names:

**Document:** [document text]

---

[2] https://openai.com/blog/chatgpt

| ID | Gold | MEGClass | X-Class | ClassKG | Document Excerpt |
|---|---|---|---|---|---|
| 1 | Britain | Britain | United States | Iraq | guests at **queen elizabeth's state banquet** were seated at an immense u shaped table brimming with floral arrangements, candelabra, fresh fruit and a dizzying array of gold plated tableware and wine glasses. the **queen** sat at the head of the table, with president bush on her right. laura bush, the first lady, sat between prince philip and prince charles, **the queen's husband and son**. karl rove, the president's top political adviser, was among the handful of americans who **took part in the royal procession**, which included the **queen's bodyguards**, wearing red jackets and carrying ceremonial halberds [...] |
| 2 | Iraq | Iraq | Britain | France | the foreign ministers of the world' s leading powers dropped diplomatic niceties, argued, shook their fingers and expressed exasperation **in a public struggle to overcome their bitter impasse on iraq**. the efforts proved futile, but they made for another day of high drama at the usually somnolent security council. "dominique, that's a false choice!" britain's foreign secretary, jack straw, declared to dominique de villepin, the french foreign minister [...] the choice everyone faced was whether to disarm **saddam hussein** peacefully or by force. [...] mr. powell was responding to the earlier presentation by hans blix , the chief inspector for biological and chemical_weapons , who had just finished saying that **iraq** was making significant strides in disarming. |
| 3 | Britain | Britain | Iraq | Iraq | **prime minister tony blair** narrowly defeated a revolt in his own **labor party** on tuesday night over **legislation in parliament to revamp the country' s higher education system**, thus avoiding a political humiliation that threatened to bring down his government. the close vote, 316 to 311 in favor of substantially raising tuition fees, **gave an important lift to mr. blair** on the eve of a potentially greater challenge to his government on wednesday, when **lord hutton** issues the findings of his investigation into the events surrounding the death of dr. david kelly. he was the specialist on iraq's weapons whose concerns, privately expressed to the bbc, formed the basis of news reports that **mr. blair and his aides** had overstated the intelligence on iraq's illicit weapons programs to make a stronger case for war. the university financing bill will require **british university students**, who like most **europeans** make only nominal tuition contributions toward the cost of a college degree, to begin paying as much as 5,500 a year starting in 2006 [...] |

Table 9: Examples excerpts of documents classified correctly by MEGClass and incorrectly by X-Class and ClassKG. Selected from the NYT-Loc dataset; we include an excerpt of the document, its ground truth (gold) label, and the corresponding predictions from each model. We bold the key indicators of the ground truth class and underline the potential red-herrings for the other models' classifications.

| Dataset | Micro-F1 | Macro-F1 |
|---|---|---|
| Yelp | 95.86 | 95.86 |
| 20News | 95.07 | 94.03 |
| NYT-Topic | 93.61 | 79.24 |
| NYT-Loc | 96.91 | 96.99 |
| Books | 78.06 | 77.17 |
| NYT-Fine | 95.77 | 87.91 |
| 20News-Fine | 79.94 | 77.30 |

Table 10: Shows accuracy of the most confident $k = 7.5\%$ of documents chosen as iterative feedback after the first iteration.

```
For the above document, select the
closest class label from ONLY the
following options: [list of class label
names]? Again, please only pick from the
provided list.
```

As demonstrated in Table 11, we find that while ChatGPT performs well on datasets with clearly distinct and coarse-grained classes (e.g. "politics" and "science" in NYT-Topic), it often misclassifies documents of type (2) in Figure 1; this consists of classes are fine-grained and their respective documents will contain keywords of similar fine-grained classes (e.g. a document under "international business" will likely contain many economic terms). This is seen through the first document in Table 11, which discusses a European carmaker shutting down a plant due to a restructuring plan as a result of a tightening budget and falling prof-

its. This event naturally requires incorporating economic terms to describe the contextual information without compromising the document-level context clearly indicating "international business" as the true class.

Similarly, the last document in Table 11 details the government shutdown regarding the repealing of the medical device tax proposed within the affordable care act. Despite the presence of federal government and budget terms at the word-level, it is clear to readers that when jointly considering word-level and document-level context, the primary subject of conversation is the affordable care act despite the similar settings in which federal budget documents take place. These examples clearly demonstrate the need for considering multiple text granularities jointly as MEGClass proposes.

### A.6 Iterative Feedback Specifications

In Table 10, we show the performance of the top $k = 7.5\%$ ranked documents chosen as iterative feedback after the first iteration. We can see that relative to the overall difficulty of each dataset, the accuracy of documents chosen is high across all classes, indicating that based on both quantitative (Table 10) and qualitative (Table 3 measures, the chosen iterative feedback is high in quality.

It is important to note that for each iteration, the top documents within the class set are up-

| ChatGPT | MEGClass/Gold | Document Excerpt |
|---|---|---|
| Economy | International Business | [...] faced with a shrinking domestic market and burning through cash, peugeot is shutting the plant at a cost of ,000 jobs, part of a restructuring plan meant to save .5 billion euros, or .1 billion, by next year. **the company says closing the plant is essential for reducing overcapacity, a problem for many european carmakers**, as plants that operate below maximum efficiency may lose money on every car they produce. **the company has five other big automotive plants in france**, and aulnay's importance had declined amid ebbing profitability for the subcompact cars it produced. [...] |
| Economy | International Business | the **european** parliament on tuesday scrapped proposals by health officials that electronic cigarettes be tightly regulated as medical devices, setting the stage for a debate in the united states over the extent of regulation . **european lawmakers endorsed a permissive approach to the sale and use of e-cigarettes**, although the products could not be sold legally to anyone younger than. the food and drug administration in the united states has said it wants to issue regulations on the nicotine-delivery devices soon. [...] |
| Federal Budget | The Affordable Care Act | with the government entering its first day of the shutdown, republicans and democrats were beginning to float ideas that they hoped would lead to a compromise."we can work on something, i believe, on the **medical device tax**," senator richard j. durbin of illinois, the no. 2 democrat, said on cnn. the tax, which helps pay for **president obama's health care overhaul**, has been a source of contention in both parties . house republicans recently included a repeal as part of a larger package of demands that the senate rejected on monday. though democrats have repeatedly insisted that they would not accept any budget deal that contains extraneous policy provisions — like the delays and defunding of **the affordable care act** that republicans have sought — mr. durbin's suggestion seemed to open the door slightly.he did have a caveat: that the revenue from the so-called **medical device tax** be replaced if repealed. at the same time, senator rand paul, republican of kentucky, suggested that congress pass a one-week budget that would allow government operations to continue while democrats and republicans talked. [...] |

Table 11: Examples of documents misclassified by ChatGPT. Selected from the NYT-Fine dataset; we include an excerpt of the document as well as the corresponding predictions from ChatGPT versus MEGClass, where MEGClass matches the ground truth (gold) class label. We bold the key indicators of the ground truth class and underline the potential red herrings.

dated/replaced, not added to (in order to avoid duplicates as well as avoid overfitting to certain documents throughout the iterations in case of any misclassifications). Furthermore, in order to avoid error propagation in case of certain document mis-classifications within our class sets, each iteration we re-learn new contextualized sentence/document representations from our initial class-oriented representations.

## A.7 Experimental Settings

For implementing **MEGClass**, we use the following hyperparameters across all datasets: $T = 100$, lr $= 1e - 3$, maximum sentence length $= 150$, number of self-attention heads $= 2$, regularization $\tau = 0.1$, epochs $= 4$, $k = 7.5\%$, and $\delta = 50\%$. For the datasets (Yelp, 20News, NYT-Topic, and NYT-Loc, 20News-Fine) with all well-defined classes (e.g. single word label names), we conduct 4 iterations of iterative feedback. For the datasets that contain phrases as label names (Books and NYT-Fine), we conduct 2 iterations. For our word representations, we use `bert-base-uncased`, which leads to the hidden dimensions $h_{cs} = 768$ and $h_{cd} = 768$.

We compare MEGClass with six other baseline text classification methods. Each baseline method is evaluated using the extremely weak supervision setting and hyperparameters as specified in their original implementation. We conduct all experiments on a single NVIDIA RTX A6000 GPU.

**- NPPrompt (Zhao et al., 2022):** We used the code-

base of NPPrompt[3]. The hyperparameters are set to the default values.

**- WDDC-MLM (Zeng et al., 2022):** We used the codebase of WDDC-MLM[4]. The hyperparameters are set to the default values. We specifically choose to compare with WDDC-MLM as opposed to their other proposed model, WDDC-Doc, where the former uses the supervision signals from a masked language model (MLM) as opposed to the document itself for the latter. We do this because WDDC-MLM has better performance than WDDC-Doc across the majority of the tested datasets.

**- ClassKG (Zhang et al., 2021):** We used the codebase of ClassKG[5]. The hyperparameters are set to the default values. We trained the graph neural network for 10 iterations.

**- X-Class (Wang et al., 2021)** We used the codebase of X-Class[6]. As before, the hyperparameters are set to the default values.

**- LOTClass (Meng et al., 2020b)** We use the codebase of LOTClass[7]. As before, the hyperparameters are set to the default values.

**- ConWea (Mekala and Shang, 2020)** We used the codebase of ConWea[8]. We utilize the class label names as seed words provided in the source code.

**- Supervised (Devlin et al., 2019)** We use the Hug-

---

[3]https://github.com/XuandongZhao/NPPrompt
[4]https://github.com/HKUST-KnowComp/WDDC
[5]https://github.com/zhanglu-cst/ClassKG
[6]https://github.com/ZihanWangKi/XClass
[7]https://github.com/yumeng5/LOTClass
[8]https://github.com/dheeraj7596/ConWea

gingFace Transformer Python Interface[9] to train the BERT model on the specified dataset using an 80-20 train-test split with the bert-base model. The text classifier is trained over 10 epochs and a batch size of 32.

## A.8  Implementation & Complexity Analysis

We detail the architecture of the multi-head self-attention (MHS) network (Section 3.3) below, which learns the contextualized sentence and document representations. The input and hidden dimensions are 768 (to preserve the original BERT embedding dimensions), and we only use two self-attention heads to minimize complexity as much as possible. We train it from scratch (no pre-training), and it has exactly 2,955,265 total parameters.

- Input: $x \leftarrow$ Initial sentence representations

- Layer 1: $x_1 \leftarrow$ MHSA(query=$x$, key=$x$, value=$x$)

- Layer 2: $x_2 \leftarrow$ LayerNorm($x_1 + x$) (contextualized sentences)

- Layer 3: $x_3 \leftarrow$ Tanh(Linear($x_2$))

- Layer 5: $w \leftarrow$ Softmax(Linear($x_3$, out-dim=1)) (sentence weights)

- Output: $cd \leftarrow w \times x_3$ (contextualized documents)

Given $N_d$ (# of documents), $N_k$ (# of classes), $N_e$ (# of epochs), $N_b$ (batch size), and $N_{CD}$ (parameters in multi-head self-attention network), we specify a time complexity analysis of MEGClass's core framework:

- Class distribution estimation: $O(N_d N_k)$

- Contextualized embeddings: $O(N_e N_b N_{CD})$

- Document Selection for Iterative Feedback: $O(N_d N_k)$

The primary time bottleneck comes from the initialization step before the core framework, where a sentence representation must be computed for each sentence, so this ultimately depends on the type of encoder used for this step (e.g. X-Class's class-oriented representations, SentenceTransformers as shown in Section A.3).

---

[9]https://github.com/huggingface/transformers/

## A.9  Evaluation Metrics

Following the prior text classification studies (Zeng et al., 2022; Zhang et al., 2021; Mekala and Shang, 2020; Wang et al., 2021; Meng et al., 2018), we evaluate our methods and the baselines using two metrics: Micro-F1 and Macro-F1.

**Macro-F1.** Macro-F1 is calculated using Macro-Precision ($P_{ma}$) and Macro-Recall ($R_{ma}$) where

$$P_{ma} = \frac{1}{|M|} \Sigma_{m \in M} \frac{|t_m \cap \hat{t_m}|}{\hat{t_m}}$$

$$R_{ma} = \frac{1}{|M|} \Sigma_{m \in M} \frac{|t_m \cap \hat{t_m}|}{t_m}$$

**Micro-F1.** Micro-F1 is calculated using Micro-Precision ($P_{mi}$) and Micro-Recall ($R_{mi}$) where

$$P_{mi} = \frac{\Sigma_{m \in M} |t_m \cap \hat{t_m}|}{\Sigma_{m \in M} \hat{t_m}}$$

$$R_{mi} = \frac{\Sigma_{m \in M} |t_m \cap \hat{t_m}|}{\Sigma_{m \in M} t_m}$$

Macro-F1 and Micro-F1 are calculated using the F1 score formula with their respective granular precision and recall scores.

$$F_1 = \frac{2 * Precision * Recall}{Precision + Recall}$$