# OpenReview forum: "MEGClass: Extremely Weakly Supervised Text Classification via Mutually-Enhancing Text Granularities"
_EMNLP/2023/Conference — EMNLP 2023 Findings_

### Official Review · Reviewer_7Hfk · 2023-08-02

**Typos Grammar Style And Presentation Improvements:** 1) Can the Figure 3 caption describe …
**Soundness:** 3

**Excitement:**

3: Ambivalent: It has merits (e.g., it reports state-of-the-art results, the idea is nice), but there are key weaknesses (e.g., it describes incremental work), and it can significantly benefit from another round of revision. However, I won't object to accepting it if my co-reviewers champion it.

**Missing References:**

1) In the Subsection "contextualized document representations," can we add some reference for regularized contrastive loss? Equation 5 looks similar to Equation 6 in [1]. If not, can you comment?

References
1) Contrastive Regularization for Semi-Supervised Learning

**Paper Topic And Main Contributions:**

The paper proposes a sample selection strategy for training text classification models using only class surface names. To select the most representative samples, the paper proposes to utilize word, sentence, and document-level context in an integrated fashion through a learned function. The paper also studies iterative feedback mechanisms to augment the informativeness of the class surface names.

**Questions For The Authors:**

1) Can you comment on the runtime performance of various methods across datasets?

2) In Section 3.3, multi-head self-attention was mentioned. Is this something custom-made or similar to the transformer architecture?

**Reasons To Accept:**

1) Idea of utilizing words, sentences, and documents level context in an integrated fashion to identify the true label of a document where only one class topic is discussed at the document level but several topics are discussed at the word or sentence level. The method seems to benefit more for longer documents.

2) Evaluation is thorough. The paper compares performance improvements in F1 scores and provides a run-time comparison of various methods.

**Reasons To Reject:**

1) No quantitative results were presented in the main paper on how the methods will perform compared to chatGPT/LLMs in a zero-shot setting in finding class representative samples. A section in the paper's appendix mostly focuses on qualitative results.

2) The paper reports results on only bert-base-uncased. Requires a study using various LLM as base embeddings to understand the impact on different base embeddings in MEGClass.

**Reproducibility:**

4: Could mostly reproduce the results, but there may be some variation because of sample variance or minor variations in their interpretation of the protocol or method.

**Reviewer Confidence:**

4: Quite sure. I tried to check the important points carefully. It's unlikely, though conceivable, that I missed something that should affect my ratings.

---

> ### Author Rebuttal · Authors · 2023-08-29
>
> We sincerely appreciate your feedback and suggestions. We are happy that you find MEGClass’s idea for integrating multiple levels of granularity, especially for more challenging documents, to be helpful as well as substantiated through our “thorough” evaluation. We have taken your suggestions seriously and have conducted additional experiments to supplement our existing ablation studies/runtime analysis and better substantiate the need for MEGClass.
>
> _No quantitative results were presented in the main paper on how the methods will perform compared to chatGPT/LLMs in a zero-shot setting in finding class representative samples. A section in the paper's appendix mostly focuses on qualitative results._
>
> - We focused on qualitative results in Section A.4 because the GPT 4 API is not open-source. ChatGPT also requires extensive prompt engineering while not guaranteeing a consistent output format nor the predicted label being within the set of provided classes. Additionally, **through our qualitative comparison in Section A.4, we demonstrate ChatGPT’s existing challenges to deal with documents that contain keywords from similar fine-grained classes** (e.g. terms from the “economy” class used in an “international business” document), where it is necessary to jointly consider the word-level and document-level context for properly understanding the primary subject of conversation– which MEGClass does.
>
> - Nonetheless, given that LLaMA-13b qualifies as a significantly large language model (as stated, 13 billion parameters) and is open source, we have presented a quantitative comparison between MEGClass and zero-shot LLaMA-13b for the full NYT-Fine corpus (only one dataset given the rebuttal time constraint). After parsing out the answer from the generated response and running evaluation, we can see that **MEGClass significantly surpasses LLaMA-13b on this dataset**. We find that the reason behind this is that LLaMA-13b tends to predict the class “football” (originally indicating American Football in the NYT-Fine corpus) for the majority of all sports classes, which represent the majority of the dataset. Additionally, it rarely selects fine-grained classes like “international business” or “law enforcement”, indicating a lack of class understanding.
>
> | Dataset/Model | MEGClass | LLaMA-13b |
> |---|---|---|
> | NYT-Fine | **89.72/71.61** | 35.25/35.59 |
>
> _The paper reports results on only bert-base-uncased. Requires a study using various LLM as base embeddings to understand the impact on different base embeddings in MEGClass._
>
> - In Section A.2, we chose to study different initializations of sentence and class representations as they are the primary input into our core framework. We specifically compare the sentence representations we use (detailed in Section 3.1; applies X-Class’s class-oriented word weighting mechanism to the bert-base-uncased embeddings) with all-mpnet-base-v2 (SentenceTransformers) and analyze this impact. The initialized sentence representation uses bert-base-uncased so that we can provide a fair comparison with the rest of the baselines, which all primarily use bert-base-uncased as their base model. Nonetheless, we include a comparison between bert-base-uncased and bert-large-uncased on the 20News, Books, and NYT-Topic datasets (just three chosen for the sake of time; the three span different document lengths, writing styles, and domains). _It is important to note that a recent paper [1] studying extremely weakly supervised methods has observed the drop in performance from bert-base and bert-large and a failure with roberta-base on X-Class’s weighing mechanism_, and they hypothesized that this may be due to the “distribution of similarities of representations generated by a language model [being] different” across models. Since we use X-Class’s class-oriented representations for sentences and their class representations, this may explain the drop in performance. Users can switch out these initial sentence representations used instead, as we tested in Section A.2, which will have a more reasonable impact on MEGClass’s performance:
>
> |  | 20News | Books | NYT-Topic |
> |---|---|---|---|
> | bert-base-uncased | **82.36/81.46** | **56.3/55.7** | **86.04/68.84** |
> | bert-large-uncased | 73.64/69.47 | 33.0/33.1 | 68.69/59.53 |
> | roberta-base | 16.28/10.89 | 7.74/5.08 | N/A |
>
> - [1] Wang, Z., Wang, T., Mekala, D., & Shang, J. (2023). A Benchmark on Extremely Weakly Supervised Text Classification: Reconcile Seed Matching and Prompting Approaches. arXiv preprint arXiv:2305.12749.
>
> _Can you comment on the runtime performance of various methods across datasets?_
>
> - Yes, across the datasets, the runtime performance is proportional to the 20News runtime analysis results (same order) and simply varies based on the number of documents and classes. We show the comparison for NYT-Fine as it has the highest number of classes (26) and hence can provide more insight into how the different methods scale with respect to the number of classes. MEGClass, LOTClass, and X-Class have their performance minimally affected, whereas ClassKG and WDDC scale in units of hours. This similar type of behavior is observed across the other datasets:
>
> | (Class #, Doc #) | MEGClass | LOTClass | X-Class | WDDC | ClassKG |
> |---|---|---|---|---|---|
> | 20News (5, 17.9K) | 1h 20m | 1h | 42m | 5h 54m | 30h |
> | NYT-Fine (25, 13.1K) | 1h 55m | 1h 21m | 1h  | 8h 19m | 42h |
>
> _In Section 3.3, multi-head self-attention was mentioned. Is this something custom-made or similar to the transformer architecture?_
>
> - We use the standard multi-head self attention block that is also used in the Transformer architecture, but it is only one block in our “custom-made”, light-weight network implementation (we define it in our source code under MEGClass/model.py). Our specific custom-made network learns the contextualized sentence representations, their corresponding sentence weights, and the contextualized document representations based on the former two. Using this custom network, we can directly update the sentence weights themselves based on our novel weighted regularized contrastive loss.
>
> _Can the Figure 3 caption describe the figure more elaborately?_
>
> - We included longer descriptions of each step within the figure itself, which covers the majority of the high-level details we discuss in Section 3.4, but we will consider how to make the figure and caption more clear!
>
> _Some parts of A.6 need to bring in Section 4.1. It was hard to figure out that bert-base-uncased was used._
>
> - We moved Experiment Setup to the Appendix for the sake of brevity, but we can move these details into Section 4.1 in the updated version since an additional page in the final version is allowed!
>
> _In the Subsection "contextualized document representations," can we add some reference for regularized contrastive loss? Equation 5 looks similar to Equation 6 in [1]. If not, can you comment?_
>
> - Yes this is a fair citation that we originally missed, we can add this reference for regularized contrastive loss. Thank you!

---

### Official Review · Reviewer_YqtT · 2023-08-08

**Soundness:** 4

**Excitement:**

4: Strong: This paper deepens the understanding of some phenomenon or lowers the barriers to an existing research direction.

**Paper Topic And Main Contributions:**

The authors propose a novel weakly supervised text classification method that utilizes various granularities, such as words, sentences, and documents. They also introduce an iterative feedback mechanism to construct pseudo-labeled training data, which is subsequently utilized for training a task-specific classifier. The paper comprehensively elucidates the problem's premise and underscores the significance of employing distinct text granularities in text classification tasks. The authors conduct experiments on several benchmark datasets, achieving state-of-the-art performance on the majority of them. Furthermore, they conduct various ablation studies to provide substantial evidence supporting the claims presented in the paper.

**Questions For The Authors:**

1. Maybe I missed the details, but I didn't get what is the source of the unlabeled dataset used. Are you using the original training data from each dataset (unlabeled)? Or is there some common general text dataset like Wikipedia to select the best sentences in the pseudo training data?
2. For Figure 4, it would be better to mention the environment used for performing the experiment.


**Reasons To Accept:**

1. The paper presents both qualitative and quantitative results to demonstrate that the proposed MEGClass method outperforms existing weak supervision approaches. The case study in the ablation section highlights the steps involved in classifying a document into one of the predefined classes.
2. The proposed MEG-Class method is well-motivated and thoroughly explained. Each level of text granularity builds upon the previous one, and they are ultimately combined iteratively to enhance the quality of the generated pseudo training data.
3. The authors conducted experiments on six different benchmark datasets, achieving significant improvements over the baselines.
4. The writing in the paper is very clear and easy to comprehend.


**Reasons To Reject:**

1. The iterative feedback mechanism is well-studied to improve the quality of labeled training data for weak supervision in text classification [1].
2. The proposed novelty of using multiple hierarchical granularity is limited.

[1] Meng, Yu, et al. "Weakly-supervised hierarchical text classification." Proceedings of the AAAI conference on artificial intelligence. Vol. 33. No. 01. 2019.

**Reproducibility:**

5: Could easily reproduce the results.

**Reviewer Confidence:**

4: Quite sure. I tried to check the important points carefully. It's unlikely, though conceivable, that I missed something that should affect my ratings.

---

> ### Author Rebuttal · Authors · 2023-08-29
>
> We genuinely appreciate your constructive and positive feedback, and are glad that you find MEGClass to be well-motivated and thoroughly explained. We hope that our response has helped clarify any confusion, and we will definitely integrate your comments into the updated version of the paper.
>
> _The iterative feedback mechanism is well-studied to improve the quality of labeled training data for weak supervision in text classification [1]._
>
> - While WeSHClass [1] does have an iterative feedback mechanism component, they use each class’s raw confident documents to _finetune_ its respective classifier (k classifiers for k classes). **What has not been previously well studied and simultaneously what MEGClass proposes** is that instead of directly using the raw documents themselves to construct a pseudo-dataset each iteration, we can use the learned _contextualized document representation_ of each confident document to directly update its respective word-based _class representation_. Not only does this prove to be successful (Figure 6, Table 10) and incorporates the idea of multiple text granularities even at the iterative feedback level, but it also avoids fine-tuning large models every single iteration and instead only once (at the very end).
>
> - [1] Meng, Yu, et al. "Weakly-supervised hierarchical text classification." Proceedings of the AAAI conference on artificial intelligence. Vol. 33. No. 01. 2019.
>
> _The proposed novelty of using multiple hierarchical granularity is limited._
>
> - From our knowledge, while previous methods do consider word- and document-level text granularities, they (1) do not consider sentence-level granularities (MEGClass does), (2) do not consider all three levels of text granularities _jointly_ (MEGClass recognizes the dependencies between all three granularities and integrates them into a learned representation through its multi-head self-attention network– jointly updating both the learned sentence weights and document representations), and (3) consider all parts of the document to be equally significant to the document-level classification task (MEGClass accounts for vague/irrelevant sentences through its novel sentence-based weighted label ensemble as described in Section 3.2 and contextualized sentence weight learning in Equation 4). **Hence, our proposed novelty is with respect to our joint granularity consideration rather than the mere inclusion of multiple text granularities.**
>
> _Maybe I missed the details, but I didn't get what is the source of the unlabeled dataset used. Are you using the original training data from each dataset (unlabeled)? Or is there some common general text dataset like Wikipedia to select the best sentences in the pseudo training data?_
>
> - Yes, we just use the original training dataset without the labels. Since we do not use any of the labels provided anyways, we choose to evaluate on all of the training documents to show that our method can perform well on a large number of documents, where 50% of the dataset is _not_ seen by the final classifier we fine-tune (Section 3.5). Also, just to clarify, we select the best _documents_ for the pseudo training dataset– this is beneficial compared to simply using sentences since this is more adaptable to the expected input being documents with multiple topics likely being discussed at the sentence and word level. MEGClass’s angle is to be able to pick out the best pseudo-training documents that confidently discuss their respective class topic, even if they contain words or sentences that overlap with other classes.
>
> _For Figure 4, it would be better to mention the environment used for performing the experiment._
>
> - **We have included the experimental settings for each baseline experiment in Section A.6.** Specifically, each baseline method is evaluated using the extremely weak supervision setting (class label names only, no additional seed words provided) and hyperparameters as specified in their original implementation. We conduct all experiments on a single NVIDIA RTX A6000 GPU. We will add the reference to Section A.6 in the figure’s caption.

---

### Official Review · Reviewer_PUtu · 2023-08-10

**Soundness:** 3

**Excitement:**

3: Ambivalent: It has merits (e.g., it reports state-of-the-art results, the idea is nice), but there are key weaknesses (e.g., it describes incremental work), and it can significantly benefit from another round of revision. However, I won't object to accepting it if my co-reviewers champion it.

**Missing References:**

- The authors should consider citing the published version of the referenced work instead of relying solely on the arXiv citation. This practice would enhance the credibility and accuracy of the references in the paper. For instance: [1] Wang, Zihan, Dheeraj Mekala, and Jingbo Shang. "X-Class: Text Classification with Extremely Weak Supervision." In Proceedings of the 2021 Conference of the North American Chapter of the Association for Computational Linguistics: Human Language Technologies, pp. 3043-3053. 2021.

**Paper Topic And Main Contributions:**

The paper presents MEGClass, a novel framework for text classification under extremely weak supervision using mutually enhancing text granularities. Unlike traditional methods relying on human annotations or class seed words, MEGClass utilizes class surface names as supervision. By jointly considering documents, sentences, and words, MEGClass captures context and inter-granularity class disagreements. It employs a Multi-Head Attention Network to create contextualized document representations, refining them through iterative feedback. Experimental results across benchmark datasets demonstrate MEGClass's superiority over other weakly and extremely weakly supervised methods, especially for longer documents and fine-grained classes. MEGClass's contributions include its innovative use of class surface names, context-aware representation learning, and the generation of a refined pseudo-training dataset.

**Questions For The Authors:**

- Could you please provide an explanation for the substantial disparity between the experimental results presented in your main study and those reported in the original paper? Addressing this discrepancy would help dispel any unwarranted doubts and provide a clearer understanding of the findings.
- It would be beneficial to elucidate the rationale behind the omission of certain State-of-the-Art (SOTA) methodologies, such as (LOPS: Learning Order Inspired Pseudo-Label Selection for Weakly Supervised Text Classification), from the comparative analysis. Similarly, clarifying the reasons for not comparing with fine-grained datasets like DBpedia and 20News would enhance the transparency of your research design.

**Reasons To Accept:**

- I find the concept of incorporating multi-granularity information (word-level, sentence-level, and document-level) for weakly supervised text classification quite intriguing. The authors effectively substantiate their approach through a thorough set of experiments, highlighting the remarkable performance of their methods.
- The authors notably create a pseudo-training dataset by estimating label distributions and iteratively refining it through feedback for fine-tuning the classifier.
- The introduction of a regularized contrastive loss to update contextualized document representations and sentence attention weights in alignment with the target class distribution is a noteworthy design choice.

**Reasons To Reject:**

- The schematic representation of the model lacks clarity and adherence to academic standards. Some components appear to be directly sourced from other papers or online without proper references. Additionally, inconsistencies in fonts and stretching of diagrams in the experimental section compromise the overall presentation.
- While the paper commences with a toy example, the motivation behind the research is not adequately articulated. This leaves a gap in understanding the precise challenges the authors aim to address in weakly supervised text classification, leading to some confusion.
- The absence of paired t-tests undermines the persuasiveness of the primary experimental results, weakening the credibility of the findings.
- The experimental evaluation should be extended to include fine-grained datasets like DBpedia and 20News, both of which possess fine-grained labels. This broader testing would enhance the paper's applicability and insights.
- Another round of proofreading is recommended to rectify minor typographical errors ('homerun') and instances of inappropriate diction, ensuring the paper maintains a high standard of clarity and professionalism.

**Reproducibility:**

4: Could mostly reproduce the results, but there may be some variation because of sample variance or minor variations in their interpretation of the protocol or method.

**Reviewer Confidence:**

4: Quite sure. I tried to check the important points carefully. It's unlikely, though conceivable, that I missed something that should affect my ratings.

---

> ### Author Rebuttal · Authors · 2023-08-29
>
> We sincerely appreciate your constructive feedback. We are encouraged that you find MEGClass’s use of multi-granularity information intriguing and that our method’s performance is “remarkable”. We have taken your constructive feedback seriously and have added a plethora of experiments for substantiating the statistical significance of our method’s SOTA performance, as well as on the datasets you have suggested. We are happily addressing your comments on our framework diagrams and are working on refining them.
>
> _While the paper commences with a toy example, the motivation behind the research is not adequately articulated. This leaves a gap in understanding the precise challenges the authors aim to address in weakly supervised text classification, leading to some confusion._
>
> - Document classification using class name only (with no training examples nor human annotations) is a challenging but useful task. This becomes even more challenging with complex/long text since a document about class A could contain many words in class B (e.g., a document on education may contain words or sentences related to politics).  Existing methods do not consider different levels of granularity and treat all sentences as equally significant for document-level classification. MEGClass automatically weighs the importance of each sentence in classification (lowers the weight of the sentences that indicate no class or multiple classes). We further developed a multi-head self-attention network to learn a document representation that reflects the critical information at multiple text granularities. Finally, we take the most confident document representations to enhance our initial word-based class representations through our iterative feedback approach, helping our model better understand what a class A document should look like at all levels of granularity. Because our multi-granularity approach can deal with these more complex documents and consider them for our final pseudo-training dataset, we allow our final classifier to be fine-tuned on more complex examples and consequently be more robust to challenging real-world documents.
>
> _The absence of paired t-tests undermines the persuasiveness of the primary experimental results, weakening the credibility of the findings._
>
> - Thank you for your feedback! In order to complete paired t-tests, we would have to run evaluation for each dataset on all of the baselines, this would require a substantial amount of time (e.g. ClassKG takes 30-40 hours). Hence, we include below the mean and variance of five trials conducted for each dataset, and we compare it with the most competitive baseline respectively. We can see that across the five trials, despite some variance (due to the randomized weight initialization when training the multi-head self-attention network and fine-tuning the final classifier), **MEGClass still reliably surpasses the SOTA for five out of six baselines, as originally stated**. We can update Table 1 with these averaged results and respective variances.
>
> |  | Test #1 | Test #2 | Test #3 | Test #4 | Test #5 | Mean | Var | Top Baseline |
> |---|---|---|---|---|---|---|---|---|
> | NYT-Topic | 86.04/68.84 | 86.04/68.48 | 85.13/67.87 | 85.21/67.77 | 84.70/67.17 | **85.42/68.03** | 0.283/0.338 | 81.67/71.54 (ConWea) |
> | NYT-Loc | 93.2/91.8 | 93.53/92.54 | 92.53/91.52 | 93.02/91.87 | 93.02/91.93 | **93.06/91.93** | 0.105/0.112 | 89.51/89.68(X-Class) |
> | Books | 56.3/55.7 | 56.40/55.46 | 56.54/55.76 | 56.33/55.70 | 56.16/55.91 | **56.35/55.71** | 0.015/0.015 | 55.0/54.7(ClassKG) |
> | NYT-Fine | 89.72/71.61 | 89.19/71.34 | 89.42/70.75 | 88.45/70.91 | 89.46/70.7 | **89.24/71.06** | 0.187/0.126 | 88.86/70.5(ClassKG) |
> | 20News | 82.36/81.46 | 81.45/80.30 | 81.60/80.61 | 81.72/80.49 | 81.46/80.3 | **81.72/80.63** | 0.113/0.185 | 81.0/82.0 (ClassKG) |
> | Yelp | 87.53/87.53 | 87.26/87.26 | 87.64/87.64 | 87.36/87.35 | 87.28/87.28 | 87.41/87.41 | 0.022/0.022 | **91.2/91.2** (ClassKG) |
>
> _Could you please provide an explanation for the substantial disparity between the experimental results presented in your main study and those reported in the original paper? Addressing this discrepancy would help dispel any unwarranted doubts and provide a clearer understanding of the findings._
>
> - Some of the compared baselines were originally designed for weakly supervised text classification (class label names + class-indicative keywords). To make the comparison fair, we modified their weak supervision to match our extremely weak supervision setting (only class label names). This led to some results for certain baselines being lower than what they reported. However, some are also higher as shown in the following table:
>
> | Baseline Model | Dataset | Their Reported F1 | Our Reported F1 |
> |---|---|---|---|
> | ConWea | 20News | 62/57 | **75.73/73.26** |
> | X-Class | NYT-Topic | **79.02**/68.55 | 79.01/**68.62** |
> | ClassKG | 20News | 80/75 | **81.0/82.0** |
>
> _It would be beneficial to elucidate the rationale behind the omission of certain State-of-the-Art (SOTA) methodologies, such as (LOPS: Learning Order Inspired Pseudo-Label Selection for Weakly Supervised Text Classification), from the comparative analysis._
>
> - LOPS is not a standalone weakly supervised text classification method, rather it enhances the performance of existing weakly supervised methods as a “performance-boost plug-in”. LOPS requires the pseudo-labels from _existing methods_ as input and additional class seed words (easier supervision level than ours) to optimize classifier fine-tuning. Integrating a performance boost plug-in into MEGClass is not within the scope of this paper, which compares the core weakly supervised text classification method with other core baselines. However, this is a promising research direction that we will explore in the future. Otherwise, to the best of our knowledge, we have included all recent and strong SOTA weakly/extremely weakly supervised text classification methods.
>
> _Similarly, clarifying the reasons for not comparing with fine-grained datasets like DBpedia and 20News would enhance the transparency of your research design._
>
> - We did not include DBPedia and 20News-Fine because not all of the baselines included them. Nonetheless, we covered sentiment-based reviews (Yelp), lengthy news articles (20News, NYT-Topic, NYT-Location), concise summaries with abstract classes (Books), and fine-grained classes (NYT-Fine). However, DBPedia explicitly defines the document’s topic in the first sentence, restricting its evaluation to word-level information and thus lacking challenging real-world scenarios with word or sentence-level content conflicting with the true document-level topic. For 20News-Fine, we favored NYT-Fine for evaluating fine-grained performance over 20News-Fine due to its training noise (e.g. gibberish documents). Still, we present MEGClass's performance on both datasets below with the original hyperparameters detailed in our paper. With the limited time, we compared MEGClass with the Macro-F1 scores of extremely weakly supervised baselines reported recently in [1]. **Notably, our method outperforms the SOTA by ~5.5 points on 20News-Fine and trails only behind ClassKG on DBPedia (Figure 4), an inefficient word-level approach.**
>
> - [1] Wang, Z., Wang, T., Mekala, D., & Shang, J. (2023). A Benchmark on Extremely Weakly Supervised Text Classification: Reconcile Seed Matching and Prompting Approaches. arXiv preprint arXiv:2305.12749.
>
> |  | LOTClass | X-Class | ClassKG | MEGClass (us) |
> |---|---|---|---|---|
> | 20News-Fine | 9.40 | 58.78$^\dagger$ | 52.29 | **64.27** |
> | DBPedia | 57.98 | 89.50 | **94.75** | 92.14$^\dagger$ |
> _$\dagger$ denotes the second-best method._
>
> _The schematic representation of the model lacks clarity and adherence to academic standards. Some components appear to be directly sourced from other papers or online without proper references. Additionally, inconsistencies in fonts and stretching of diagrams in the experimental section compromise the overall presentation._
>
> - Thank you for your feedback, we appreciate the valuable comments regarding our presentation. We will fix the inconsistent font formatting of diagrams and minor typographical errors in the final version of the paper. We will work on making the main schematic representation much clearer in the updated version and have also replaced the diagram used for the contextualized sentences component but nonetheless added the Transformer paper's citation to our references list, thank you for pointing this out!

---

### Official Review · Reviewer_pjys · 2023-08-11

**Soundness:** 3

**Excitement:**

4: Strong: This paper deepens the understanding of some phenomenon or lowers the barriers to an existing research direction.

**Paper Topic And Main Contributions:**

The authors of this paper propose a pipeline for learning from weak supervision signals of different granularities. They distinguish between document- and sentence-/word-level, both of which are taken into account in their approach, whereas only the document-level was taken into account in previous research. In their work, they exploit the information on different granularities to refine both the sentence and the document representations, as well as the word-based class representations via iterative feedback loops. While surpassing the compared weakly-supervised baselines on 5 out of 6 benchmarks, there still remains a notable gap to the fully supervised baseline.

**Questions For The Authors:**

A: l. 298: How do you justify to only look at the top 2 classes? Out of simplicity? Did you conduct ablation studies wrt. to this choice? What if a sentence pretty well discriminates e.g. two classes from the rest but is unable to distinguish among them?
B: The overall model size/complexity does not become very clear. I understand that you use BERT as a classifier, but what are the specifications of the whole architecture before BERT?



**Reasons To Accept:**

- This is a practically very relevant scenario for practitioners and there is a lack of existing benchmark data sets for evaluating a model’s capabilities with respect to solving this task.
- The method is innovative and intuitively understandable
- The presentation of the paper is overall very clear and easy to follow
- The analysis is very thorough; the choice of data sets, performance metrics, and ablations are meaningful and well justified



**Reasons To Reject:**

- The pipeline consists of many components for each of which it does not become entirely clear how high the contribution to the overall performance actually is. So it feels kind of handcrafted and potentially feels somehow brittle since the only hyperparameters that are more closely inspected are the temperature and #epochs.
- The overall model size / complexity does not become very clear.

**Reproducibility:**

5: Could easily reproduce the results.

**Reviewer Confidence:**

3: Pretty sure, but there's a chance I missed something. Although I have a good feel for this area in general, I did not carefully check the paper's details, e.g., the math, experimental design, or novelty.

**Typos Grammar Style And Presentation Improvements:**

l. 14/15, l. 44, l. 64 (and other places): Write weakly-supervised either w/ or w/o hyphen
l. 472 afaik the data set is called “20newsgroups”
Fig. 5 is not appropriate; the cut-off y-axis conveys a misleading message as differences between the different bars are grossly overemphasized.

---

> ### Author Rebuttal · Authors · 2023-08-29
>
> We greatly appreciate your constructive and positive comments and helpful questions. We are thankful that you found MEGClass to be very relevant for practitioners, innovative, and intuitively understandable. We have taken your questions very seriously, and hopefully our response provides more clarity on our method’s systematic approach and time complexity.
>
> _So it feels kind of handcrafted and potentially feels somehow brittle since the only hyperparameters that are more closely inspected are the temperature and #epochs._
>
> - We introduce four core hyperparameters in MEGClass, and to clarify, we conducted hyperparameter sensitivity analyses for each one in the original paper. At the high-level, our framework consists of three main parts: (1) initialization (Section 3.1), (2) pseudo-training dataset refinement (Sections 3.2-3.4), and (3) classifier fine-tuning (Section 3.5). The second part is our main component, which involves the following three substeps:
>
>      i. Class distribution estimation (Section 3.2) → there are **no hyperparameters** for estimating the initial class distribution of each document.
>
>      ii. Contextualized embeddings (Section 3.3) → for learning document representations that jointly integrate sentence and document level information), the hyperparameters we introduce are the **training epochs and regularization temperature** used in the weighted regularized contrastive loss. **Figures 7 and 8 demonstrate their general stability and that our selected values for both have stable performance across six diverse datasets.**
>
>      iii. Iterative feedback for refining the first two substeps (Section 3.4) → we introduce the following hyperparameters: (1) k for choosing the **top k%** contextualized document representations to enhance each class representation, and (2) the **number of iterations** of feedback. **Figure 6 presents a sensitivity analysis on both of these, and shows that we reach a relatively stable set of confident documents at the same k% across all six datasets.**
>
> - **For a fair comparison with the baselines, we use the same hyperparameter settings for the initialization and classifier fine-tuning steps** (e.g. for the final classifier finetuning, choosing the top $\delta=50$% of documents as the pseudo-training dataset).
>
> _The pipeline consists of many components for each of which it does not become entirely clear how high the contribution to the overall performance actually is._
>
> - In our thorough evaluation, we analyzed the relative contributions of each component and found that the impact of our learned contextualized representations (ii) > iterative feedback (iii) > class distribution estimation (i). Specifically, the quality of the joint representations that we learn determines which documents we deem as sufficiently confident to enhance the class representations as iterative feedback, which then heavily influences the accuracy of the estimated class distribution used for the next iteration. We can see this immediate boost in performance from (ii) and (iii) in Figure 6, as well as the overall high quality documents chosen across all different datasets, shown in Table 10. We can make this explicit comparison between components more apparent in our existing “ablation studies” subsection under Section 4.2.
>
> _The overall model size/complexity does not become very clear. I understand that you use BERT as a classifier, but what are the specifications of the whole architecture before BERT?_
>
> - Following our previous response, the architecture of substep (ii), the multi-head self-attention network that learns the contextualized sentence and document representations, is:
>
>      - Input: $x$ ← Initial sentence representations
>      - Layer 1: $x_1$ ← MultiHeadSelfAttention(query=$x$, key=$x$, value=$x$)
>      - Layer 2: $x_2$ ← LayerNorm($x_1$ + $x$) (contextualized sentences)
>      - Layer 3: $x_3$ ← Tanh(Linear($x_2$))
>      - Layer 5: $w$ ← Softmax(Linear($x_3$, out-dim=1)) (sentence weights)
>      - Output: $cd$ ← $w \bigtimes x_3$ (contextualized documents)
>
> - The input and hidden dimensions are 768 (to preserve the original BERT embedding dimensions), and we only use two self-attention heads to minimize complexity as much as possible. We train it from scratch (no pre-training), and it has exactly 2,955,265 total parameters.
>
> - Given $N_d$ (# of documents), $N_k$ (# of classes), $N_e$ (# of epochs), $N_b$ (batch size), and $N_{CD}$ (parameters in multi-head self-attention network), here is a time complexity analysis of MEGClass’s core framework:
>
>      i. Class distribution estimation: $O(N_dN_k)$
>
>      ii. Contextualized embeddings: $O(N_eN_bN_{CD})$
>
>      iii. Document Selection for Iterative Feedback: $O(N_dN_k)$
>
> - The primary time bottleneck comes from the initialization step before the core framework, where a sentence representation must be computed for each sentence, so this ultimately depends on the type of encoder used for this step (e.g. X-Class’s class-oriented representations, SentenceTransformers as shown in Section A.2). We can add this analysis to the Appendix!
>
> _How do you justify to only look at the top 2 classes? Out of simplicity? Did you conduct ablation studies wrt. to this choice? What if a sentence pretty well discriminates e.g. two classes from the rest but is unable to distinguish among them?_
>
> - We examine the _distinction_ between the top two classes; if the top two classes are equally strong, their distinction is 0, and it has no value for single-label classification. For example, a document which maps 50% to economy, 50% to international business provides no value for single-label text classification. On the other hand, a _<70%, 30%> split provides good value for single-label classification_. As shown in Figure 5, our ablation studies demonstrate the superiority of our sentence weight computation based on these top two classes over other common weighing mechanisms. We do acknowledge that your point of taking into account more than two classes would be very beneficial if we wanted to extend our setting to the multi-label classification. This could be a topic for future work!

---

### Meta-Review · Area_Chair_x4Sf · 2023-09-19

**Recommendation:** 3

**Metareview:**

The reviewers all recognized the importance of this task for practitioners and are excited about how intuitive the presented methods are, and I agree that this is an important and interesting task. This could be even better supported, however, as one reviewer raised a concern about the motivation of the paper and specifically the toy example given in Section 1. I also found this hard to follow; perhaps the authors can select a more intuitive and/or compelling example, which would address this concern. Generally though, the paper is very well motivated and this is recognized by all the reviewers.

Multiple reviewers expressed confusion over the system explanation and visualization. This part of the paper should be clarified and additional context should be given about the sources of the various components. I found this part confusing as well.

Reviewers gave ambiguous soundness scores, but I believe these concerns were mostly addressed in the rebuttal (e.g., the variation tests in response to Reviewer PUtu).  I agree these results should have been included in the original paper, but they do address this particular concern. The authors also provided an analysis of additional datasets, on top of the many datasets already included in the submitted paper.

---

### Decision · Program_Chairs · 2023-10-07

**Decision:**

Accept-Findings

**Comment:**

The reviewers all recognized the importance of this task for practitioners and are excited about how intuitive the presented methods are, and I agree that this is an important and interesting task. This could be even better supported, however, as one reviewer raised a concern about the motivation of the paper and specifically the toy example given in Section 1. I also found this hard to follow; perhaps the authors can select a more intuitive and/or compelling example, which would address this concern. Generally though, the paper is very well motivated and this is recognized by all the reviewers.

Multiple reviewers expressed confusion over the system explanation and visualization. This part of the paper should be clarified and additional context should be given about the sources of the various components. I found this part confusing as well.

Reviewers gave ambiguous soundness scores, but I believe these concerns were mostly addressed in the rebuttal (e.g., the variation tests in response to Reviewer PUtu).  I agree these results should have been included in the original paper, but they do address this particular concern. The authors also provided an analysis of additional datasets, on top of the many datasets already included in the submitted paper.